# Elevated interleukin-8 expression by skin fibroblasts as a potential contributor to pain in women with Fabry disease

**Lukas Hofmann**[1], **Julia Grüner**[1], **Katharina Klug**[1], **Maximilian Breyer**[1], **Thomas Klein**[1], **Vanessa Hochheimer**[1], **Laura Wagenhäuser**[1], **Erhard Wischmeyer**[2¤], **Nurcan Üçeyler**[1,3]*

**1** Department of Neurology, University Hospital of Würzburg, Würzburg, Germany, **2** Molecular Electrophysiology, Institute of Physiology, Center of Mental Health, University of Würzburg, Würzburg, Germany, **3** Fabry Center for interdisciplinary Therapy (FAZiT), University Hospital of Würzburg, Würzburg, Germany

¤ Current address: Bielefeld University, Medical Faculty Ostwestfalen-Lippe, Cellular Neurophysiology, Bielefeld, Germany
* ueceyler_n@ukw.de

**Data Availability Statement:** All relevant data are within the manuscript and its Supporting Information files.

## Abstract

Fabry disease (FD) is a lysosomal storage disorder of X-linked inheritance. Mutations in the α-galactosidase A gene lead to cellular globotriaosylceramide (Gb3) depositions and trig-gerable acral burning pain in both sexes as an early FD symptom of unknown pathophysiology. We aimed at elucidating the link between skin cells and nociceptor sensitization contributing to FD pain in a sex-associated manner. We used cultured keratinocytes and fibroblasts of 27 adult FD patients and 20 healthy controls. Epidermal keratinocytes and dermal fibroblasts were cultured and immunoreacted to evaluate Gb3 load. Gene expression analysis of pain-related ion channels and pro-inflammatory cytokines was performed in dermal fibroblasts. We further investigated electrophysiological properties of induced pluripotent stem cell (iPSC) derived sensory-like neurons of a man with FD and a healthy man and incubated the cells with interleukin 8 (IL-8) or fibroblast supernatant as an *in vitro* model system. Keratinocytes displayed no intracellular, but membrane-bound Gb3 deposits. In contrast, fibroblasts showed intracellular Gb3 and revealed higher gene expression of potassium intermediate/small conductance calcium-activated potassium channel 3.1 (KCa 3.1, *KCNN4*) in both, men and women with FD compared to controls. Additionally, cytokine expression analysis showed increased IL-8 RNA levels only in female FD fibroblasts. Patch-clamp studies revealed reduced rheobase currents for both iPSC neuron cell lines incubated with IL-8 or fibroblast supernatant of women with FD. We conclude that Gb3 deposition in female FD patient skin fibroblasts may lead to increased KCa3.1 activity and IL-8 secretion. This may result in cutaneous nociceptor sensitization as a potential mechanism contributing to a sex-associated FD pain phenotype.

## Introduction

The inherited disorder Fabry disease (FD) is caused by several hundred mutations in the gene encoding the enzyme α-galactosidase A (*GLA*) [1]. Reduction or absence of GLA activity

**Funding:** Deutsche Forschungsgemeinscahft (DFG): UE171/6-1 (N.Ü.); CRC1158 (subproject A10, N.Ü.); UE171/15-1 (N.Ü.). The sponsors or funders did not play any role in the study design, data collection and analysis, decision to publish, or preparation of the manuscript.

**Competing interests:** The authors have declared that no competing interests exist.

results in a multi-organ disease with limited life expectancy and lysosomal depositions of globotriaosylceramide (Gb3) [2]. Although FD is X-linked, both sexes may reach all levels of disease severity. One of the earliest symptoms of FD is episodic acral burning pain [3] which is typically triggered by heat, inflammation or physical activity [4] and is mostly attributed to small nerve fiber impairment. Varying FD pain phenotypes reduce patients'health-related quality of life, while the pathophysiology remains unknown. Gb3 deposits in dorsal root ganglion (DRG) neurons of FD patients [5] and FD animal models [6–8] provide hints towards a link between neuronal sphingolipid accumulation and FD-associated pain. In contrast, the recently reported association of Gb3 accumulation in fibroblasts and FD-associated pain in men with FD [9] suggests another potential mechanism contributing to pain. Skin fibroblasts and keratinocytes are key players in the processing of chemical and thermal stimuli and cutaneous nociception [10–13]. Additionally, dermal fibroblasts also express pain-associated ion channels [14–16] and are linked with pro-inflammatory cytokine release in patients with small fiber neuropathy [17]. Further research underscores the role of fibroblast control of nociception in animal and *in vitro* studies [18–20]. Mechanisms of potential cross-talk of both pathways and sex-specificity remain elusive. We aimed to unravel Gb3-dependent cutaneous mechanisms potentially underlying diversity of FD pain phenotypes. We report on higher gene expression of calcium-activated potassium channel 3.1 ($K_{Ca}$ 3.1, KCNN4) and interleukin-8 by dermal fibroblasts of women with FD compared to those of men as potential contributors to sex-associated pain diversity in FD.

## Materials and methods

### Study population

Between 01.09.2015 and 31.12.2018, we prospectively recruited 27 patients with FD, 13 men and 14 women, with a median age of 46 years (range 19–74) at the Fabry Center for interdisciplinary Therapy (FAZiT), University Hospital of Würzburg, Germany. All patients underwent neurological examination, nerve conduction studies, and quantitative sensory testing (QST). We included patients, if FD was genetically confirmed and if patients showed ≥2 of the following three symptoms or signs: FD-associated pain (in childhood and/or adulthood), and/or paradoxical heat sensation during QST indicating small nerve fiber impairment, and/or reduced intraepidermal nerve fiber density (IENFD) in distal and/or proximal skin punch biopsies.

Controls were recruited at the Department of Neurology, University Hospital of Würzburg from among patients'healthy (i.e. no *GLA* variant) family members and friends. We enrolled 20 controls (14 women, six men) with a median age of 49 years (range 22–66). Our study was approved by the Würzburg Medical School Ethics Committee (#135/15) and participants were enrolled upon oral and written informed consent.

### Skin punch biopsy

Six-mm skin punch biopsies were obtained as described previously [9]. For histological determination of the IENFD and for cultivation of dermal and epidermal cells, skin samples were divided into two pieces and processed as follows.

**IENFD determination.** One half of the skin sample was used to histologically determine IENFD as described previously [9]. Briefly, skin was fixed in 4% buffered paraformaldehyde (PFA; pH 7.4), embedded in Tissue Tek® (Sakura Finetek Europe B.V., Alphen aan den Rijn, Netherlands), and stored at -80˚C until further processing. Samples were immunoreacted with antibodies against the neuronal marker protein-gene product 9.5 (PGP 9.5; 1:1000, Zytomed, Berlin, Germany) with goat anti-rabbit IgG labeled with fluorescent cyanine 3.18 (1:100; Dianova, Hamburg, Germany), for visualization of 40-µm cryosections using a Zeiss Axiophot 2

microscope (Axiophot2, Zeiss, Oberkochen, Germany) with a CCD camera (Visitron Systems, Tuchheim, Germany) and SPOT advanced software (Windows version 4.5, Diagnostic Instruments, Inc., Sterling Heights, MI, USA). We analyzed three biopsy sections per study participant each in a blinded manner. Intraepidermal nerve fibers were quantified according to published rules [21] and the mean intraepidermal nerve fiber density (i.e. the number of fibers/mm) was determined by counting intraepidermal nerve fibers, if they crossed or originated at the dermal-epidermal border. The number of fibers was divided by the length of the respective epidermis. Secondary branches and fiber fragments were spared. IENFD data were compared with our laboratory's normative data base grounding on 180 healthy controls (124 women; median age: 50 years, 20–84; 56 men; median age 53 years, 22–76) in skin biopsies from the lower leg (women: n = 109; men: n = 46) and the upper thigh (women: n = 102; men: n = 31). All skin biopsies were processed and assessed in our laboratory.

**Human skin cell culture.** Skin fibroblast cultures were prepared from the second half of the 6-mm skin punch biopsy as described earlier [22]. In brief, epidermis and dermis were separated mechanically and placed in cell culture medium until cells started to migrate into T25 culture flasks. For human dermal fibroblast (hDF) cultivation, complete Dulbecco's Modified Eagle Medium was used: Nutrient Mixture F-12, supplemented with penicillin/streptomycin (1%) and fetal calf serum (10%) (DMEM/F-12, Life Technologies, Carlsbad, CA, USA). Human epidermal keratinocytes (hEK) were cultivated in EpiLife medium (Thermo Fisher Scientific, Waltham, MA, USA) supplemented with (1%) penicillin/streptomycin and 1X Epi-Life Defined Growth Supplement (Life Technologies, Carlsbad, CA, USA). Cells were grown at 37˚C, 5% $CO_2$ (v/v) under regular control and medium change twice a week. For passaging, fibroblasts were incubated (1 min at 37˚C, 5% $CO_2$ (v/v)) with TrypLE Express (Life Technologies, Carlsbad, CA, USA) once 90% confluence was reached. Cells were stored in liquid nitrogen before further processing and were thawed and cultivated in 6-well plates, or T25 culture flasks until 80% confluence was reached. Afterwards, cells were passaged as described earlier [22]. We used maximum two passages after thawing.

## Immunocytochemistry

For immunocytochemistry, cultured hEK and hDF were fixed for 15 min in 4% PFA/phosphate buffered saline (PBS) (Merck Millipore, Billerica, MA, USA), blocked with 10% bovine serum albumin and 0.1% saponin (both, Sigma-Aldrich, St. Louis, MS, USA) for 30 min, and incubated with the following primary antibody and toxin mix over night at 4˚C: cytokeratin 10 (1:300; mouse monoclonal anti human, Abcam, Cambridge, UK) for hEK, α-tubulin (1:200; rabbit polyclonal ant human, Cell Signaling Technology, Danvers, MA, USA) for both cell types, and Shiga toxin subunit B (Stx, 1:300; Sigma-Aldrich, St. Louis, MS, USA) custom conjugated either with Alexa Fluor 647 or 555 for visualization of Gb3 [23]. On the second day, samples were incubated with fluorophore-conjugated species-specific secondary antibodies (Jackson ImmunoResearch, West Grove, PA, USA) for 30 min at room temperature, counterstained with DAPI (1:10.000, Vector Laboratories, Burlingame, CA, USA), and mounted with Aqua-Poly/Mount (Polysciences, Warrington, PA, USA).

Immunoreaction against membrane-bound Gb3 was performed on cultured hEK cells. For this, culture medium was removed and cells were rinsed with PBS. Cells were incubated with Stx custom conjugated with Alexa Fluor 647 (1:300, Sigma-Aldrich, St. Louis, MS, USA) for 30 min on ice, afterwards washed with PBS (Merck Millipore, Billerica, MA, USA), and fixed with 4% PFA/PBS (Merck Millipore, Billerica, MA, USA) for 15 min at room temperature. After the fixation step, cells were washed three times with PBS for five minutes, counterstained with DAPI, and mounted with Aqua-Poly/Mount (Polysciences, Warrington, PA, USA).

### Gene expression analysis

**mRNA preparation.**    For fibroblast detachment and lysis, QIAzol Lysis Reagent® (Qiagen, Hilden, Germany) was used. Applying miRNeasy Mini Kit (Qiagen, Hilden, Germany) and a Centrifuge 5417R (Eppendorf, Hamburg, Germany), we extracted mRNA from lysed cells including DNAse treatment. mRNA samples were stored at -80°C until further processing. We used the NanoDrop™ One (Thermo Fisher Scientific, Waltham, MA, USA) for assessment of RNA quality and quantity.

**Reverse transcription PCR.**    All reagents and cyclers used were purchased from Thermo Fisher Scientific (Waltham, MA, USA), if not reported otherwise. Applying TaqMan Reverse Transcription reagents, 250 ng of mRNA was transcribed to cDNA. Mastermix contained 5 μl random hexamer, 10 μl 10x buffer, 22 μl MgCl$_2$, 20 μl dNTP, 2 μl RNAse inhibitor, 6.2 μl multiscribe RT, and 2 μl Oligo-D and 3.5 μl of cDNA was individually added. All samples were centrifuged prior transcription. cDNA reaction was performed in a Peqlab Advanced Primus 96® cycler (PEQLAB, Erlangen, Germany) at the following conditions: 25°C, 10 min; 48°C, 60 min; 95°C, 5 min.

**Quantitative real-time-PCR (qRT-PCR).**    We purchased all reagents and cyclers from Thermo Fisher Scientific (Waltham, MA, USA). Gene expression analysis was performed for the following ion channels: transient receptor potential vanilloid 1 (*TRPV1*, Hs00218912_m1), hyperpolarization-activated cyclic nucleotide-gated ion channel 2 (*HCN2*, Hs00609603_m1), voltage-gated sodium channel 1.7 (*SCN9A*, Hs00161567_m1), voltage-gated sodium channel 1.8 (*SCN10A*, Hs01045137_m1), potassium calcium-activated channel subfamily M alpha 1 (*KCNMA1*, Hs01119504_m1), potassium intermediate/small conductance calcium-activated channel, subfamily N, member 4 (*KCNN4*, Hs01069779_m1). Additionally, we assessed gene expression of the inflammatory cytokines tumor necrosis factor-α (*TNF*, Hs00174128_m1), *interleukin-1β* (*IL-1β*, Hs00174097_m1), *IL-6* (Hs00174131_m1), and *IL-8* (Hs00174103_m1). We performed duplex PCR for target RNA and 60S ribosomal protein L13a (*RPL13A*, Hs04194366_g1) as endogenous control. For this, each well contained 5 μl Fast Advanced Mastermix (2x), 0.5 μl TaqManAssay (20x) target primer (FAM-MGB), 0.5 μl TaqManAssay (20x) endogenous control primer (VIC-MGB), 0.5 μl nuclease-free water and 8.75 ng/well cDNA adding nuclease-free water to a final volume of 10 μl. Each plate contained a negative control, a white blood cell TNF control for interplate comparability, and a calibrator sample. The calibrator for each primer was determined by calculating the control sample with the threshold cycle (Ct) value closest to the mean Ct value. All plates were run under the following conditions: 50°C, 1 min; 95°C, 12 min; 40 cycles 95°C, 3 sec and 60°C, 30 sec. Samples were measured in triplicates. All plates were analyzed with the QuantStudio™ Design & Analysis Software Version 1.5.1 (Thermo Fisher Scientific, Waltham, MA, USA) and data evaluation was performed with the ΔΔCt method [24].

### Generation of induced pluripotent stem cell (iPSC) derived neurons

Sensory-like neurons of two FD patients, one with pain (FD-1) and one without pain (FD-2), and a healthy male control (Ctrl) were generated and characterized as described elsewhere [25]. In brief, iPSC were generated from hDF using the StemRNA 3rd Gen reprogramming kit (Reprocell, Beltsville, MD, USA) following the manufacturer's instructions. Sensory-like neurons were cultivated according to a published protocol with minor modifications [26] and were matured for at least five weeks before analysis.

### Patch-clamp analysis

Whole-cell patch-clamp analysis on iPSC-derived sensory-like neurons was performed on single cells. Bath solution consisted of 180 mM NaCl, 5.4 mM KCl, 1.8 mM CaCl$_2$, 1 mM MgCl$_2$,

10 mM glucose, and 5 mM HEPES; the pH was adjusted to 7.4 [27]. Borosilicate glass capillaries were used to produce patch pipettes (Kimble Chase Life Science and Research Products, Meiningen, Germany). Pipettes were heat-polished to reach an input resistance of 1 to 4 MΩ (whole-cell). The pipette recording solution contained 170 mM KCl, 2 mM MgCl$_2$, 1 mM EGTA, 1 mM ATP, and 5 mM HEPES. Both solutions were adjusted to 290 mOsm. An EPC10 patch-clamp amplifier (HEKA, Ludwigshafen, Germany) was used to record currents with a sampling rate of 20 kHz. Stimulation and data acquisition were controlled by the Patchmaster software package (HEKA, Lambrecht, Germany) on a windows computer, and data analysis was performed off-line with GraphPad PRISM Version 8.01 (GraphPad Software, Inc., La Jolla, CA, USA).

iPSC-derived sensory-like neurons of a FD patient (male, 28 years, reporting FD pain attacks), and a healthy Ctrl (male, 60 years, no pain) were incubated with IL-8 (100 ng/ml, Peprotech, Hamburg, Germany) or hDF supernatant (1:1 iPSC neuron culture medium and hDF medium) for 1h at 37˚C. HDF supernatant was collected 24h after replacement of hDF medium with pure (serum-free) DMEM F12 from 80% confluent fibroblasts. Cells were incubated with female FD patient hDF (PatF), male FD patient hDF (PatM), both with a FD pain phenotype, and Ctrl (Ctrl) supernatant from healthy controls. Fibroblast supernatant was collected from cells of our investigated patient cohort. Supernatants were not pooled and all patient conditioned media samples were used as individual replicates.

For current clamp recordings, square current injections were applied to induce action potentials (AP; from 0 to 300 pA in 10 pA steps). For calculation of AP properties, the measurement function of the Patchmaster software package (HEKA, Lambrecht, Germany) was used and values of the first evoked AP were exported to an excel file for further processing and storage. We measured threshold potentials at the turning point of the overshoot and evaluated AP amplitudes as the difference between membrane potential and AP peak. AP duration was measured as the distance between depolarization and repolarization at the point of half maximum AP amplitude. The hyperpolarization amplitude was defined as the minimum potential after the AP. Rheobase was defined as minimum current to elicit the first AP. For all patch-clamp experiments and all conditions investigated, ≥6 cells were recorded.

## Statistical analysis

SPSS 25 (IBM, Ehningen, Germany) was used for statistical analysis and GraphPad PRISM Version 8.01 (GraphPad Software, Inc., La Jolla, CA, USA) was applied for graph preparation. Kolmogorov-Smirnov test was used to test data distribution. For non-parametric gene expression analysis of skin cells, we used Kruskal-Wallis test, followed by Dunn's multiple comparison test. Normally distributed electrophysiological parameters were analyzed using two-way ANOVAs with cell line (Ctrl, FD) and treatment (naïve, Ctrl, PatF, PatM, IL-8) as independent variables. For multiple comparison, Dunnett's test was applied. $P < 0.05$ was assumed statistically significant.

## Results

### Clinical characteristics for study inclusion

FD-associated pain was reported by 10/14 (71%) women (1x only in childhood) and 10/13 (77%) men. In 5/14 (36%) women and 3/13 (23%) men with FD, paradoxical heat sensation was detected during QST. Reduced distal IENFD was found in 9/14 (64%) women and 11/13 (85%) men with FD. Further clinical details are provided in S1 Table.

## FD keratinocytes do not show intracellular, but membrane-bound Gb3

We first investigated FD and control cells for qualitative comparison. No Gb3 depositions were detected in the cytoplasm of hEK of FD patients (Fig 1A' and 1A") and in the cytoplasm or cell membrane of control hEK (Fig 1B,), but the immunoreaction for membrane-bound Gb3 was positive in FD hEK (Fig 1B"). As expected, only few Gb3 deposits were detected in control hDF reflecting its physiological generation (Fig 1C'). In contrast, dense intracellular Gb3 deposits were visible in fibroblasts of FD patients (Fig 1C").

## Altered gene expression of pain-related ion channels in hDF of FD patients

Due to the lack of intracellular Gb3 deposits in hEK, we continued our experiments with hDF. Gene expression analysis of hDF revealed less *TRPV1* and *SCN9A* expression in FD hDF compared to Ctrl (p<0.001, Fig 2A', p<0.01; Fig 2A"), while there was no intergroup difference for HCN2 and KCNMA1 expression (Fig 2A''' and 2A''''). *KCNN4* gene expression was higher in hDF of FD patients compared to Ctrl (p<0.001, Fig 2B). When stratifying data for sex, *KCNN4* gene expression was higher in men (p<0.05) and women with FD (p<0.001) compared to Ctrl (Fig 2C). *SCN10*A and *TRPM8* were not expressed (Ct values >33 each).

## Higher inflammatory cytokine gene expression in hDF of female FD patients than in male patients and Ctrl

We found higher IL-8 gene expression in hDF of FD patients compared to Ctrl (p<0.001, Fig 3A). Performing gene expression analysis of male and female patient subgroups, higher IL-8

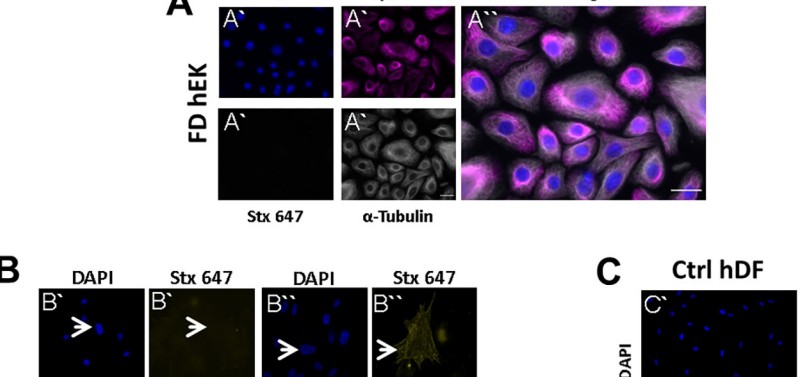

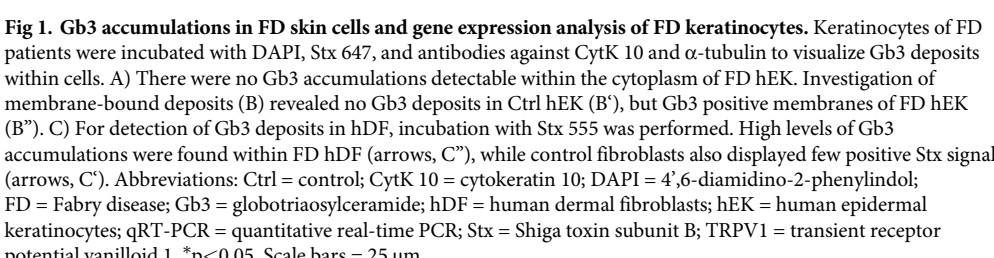

**Fig 1. Gb3 accumulations in FD skin cells and gene expression analysis of FD keratinocytes.** Keratinocytes of FD patients were incubated with DAPI, Stx 647, and antibodies against CytK 10 and α-tubulin to visualize Gb3 deposits within cells. A) There were no Gb3 accumulations detectable within the cytoplasm of FD hEK. Investigation of membrane-bound deposits (B) revealed no Gb3 deposits in Ctrl hEK (B'), but Gb3 positive membranes of FD hEK (B"). C) For detection of Gb3 deposits in hDF, incubation with Stx 555 was performed. High levels of Gb3 accumulations were found within FD hDF (arrows, C"), while control fibroblasts also displayed few positive Stx signals (arrows, C'). Abbreviations: Ctrl = control; CytK 10 = cytokeratin 10; DAPI = 4',6-diamidino-2-phenylindol; FD = Fabry disease; Gb3 = globotriaosylceramide; hDF = human dermal fibroblasts; hEK = human epidermal keratinocytes; qRT-PCR = quantitative real-time PCR; Stx = Shiga toxin subunit B; TRPV1 = transient receptor potential vanilloid 1. *p<0.05. Scale bars = 25 μm.

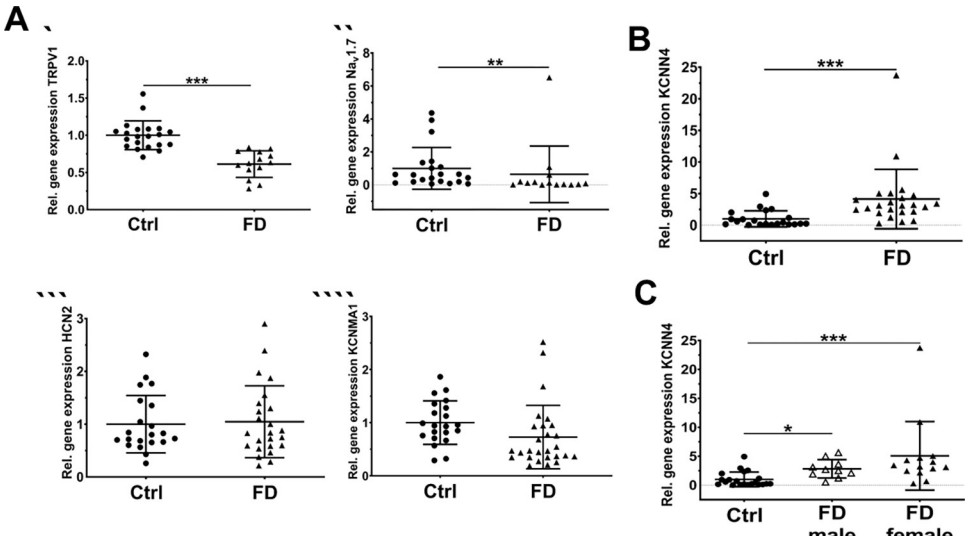

**Fig 2. Ion channel gene expression analysis of FD fibroblasts.** Pain related ion channel gene expression was investigated in hDF using qRT-PCR. Expression of *TRPV1* (p<0.001, A') and *SCN9A* (p < .01, A") was reduced compared to Ctl hDF, while there was no difference for *HCN2* (A'''), and *KCNMA1* (A'''') expression. B) Whole patient cohort hDF *KCNN4* expression was higher compared to Ctrl (p<0.001). C) Sex stratification of the patient cohort revealed a higher *KCNN4* gene expression in male (p<0.05), and female FD hDF (p<0.001). Please see supplementary file for raw data. Abbreviations: Ctrl = control; FD = Fabry disease; HCN2 = hyperpolarization-activated cyclic nucleotide-gated ion channel 2; hDF = human dermal fibroblasts; qRT-PCR = quantitative real-time PCR; TRPV1 = transient receptor potential vanilloid 1; KCNMA1 = potassium calcium-activated channel subfamily M alpha 1; KCNN4 = calcium-activated potassium channel 3.1; SCN9A = voltage-gated sodium channel 1.7. *p<0.05, **p<0.01, ***p<0.001.

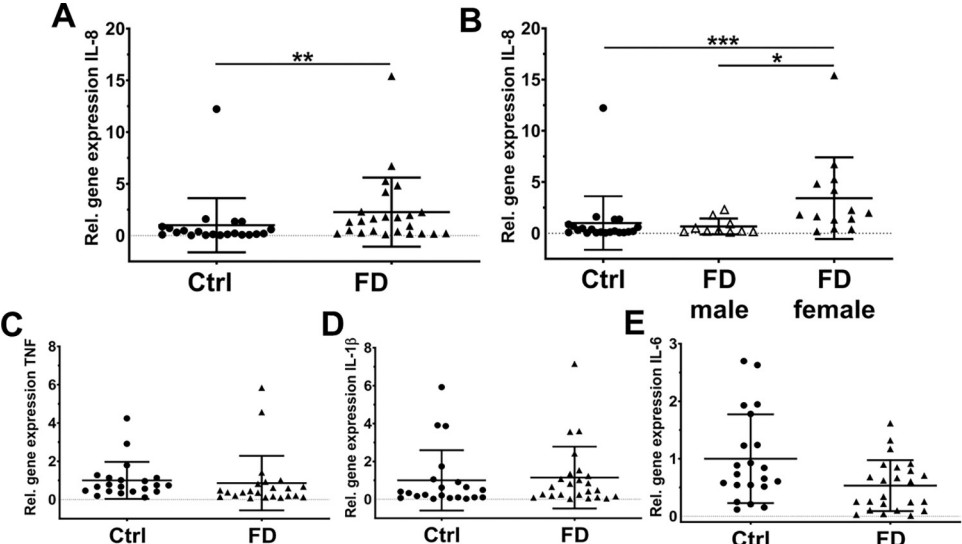

**Fig 3. Cytokine gene expression analysis of FD fibroblasts.** To investigate cytokine gene expression in hDF, qRT-PCR was used. Gene expression of *IL-8* was higher in whole FD patient fibroblasts, compared to Ctrl (p<0.01, A). Analysis of data stratifying patient groups for sex revealed higher *IL-8* gene expression only for female hDF compared to Ctrl (p<0.001) and male FD hDF (p<0.05, B). Gene expression of *TNF* (C), *IL-1β* (D), and *IL-6* (E) was not different between FD patients and Ctrl. Please see supplementary file for raw data. Abbreviations: Ctrl = control; FD = Fabry disease; hDF = human dermal fibroblasts; IL-1β = interleukin 1β; IL-6 = interleukin 6; IL-8 = interleukin 8; qRT-PCR = quantitative real-time PCR; TNF = tumor necrosis factor-alpha. *p<0.05, **p<0.01, ***p<0.001.

expression was confirmed only for hDF of women with FD compared to hDF of men ($p < 0.05$) and Ctrl ($p < 0.001$, Fig 3B). There was no intergroup difference for gene expression of TNF, IL-1β, and IL-6 (Fig 3C–3E).

## FD hDF supernatant leads to reduced AP thresholds

AP parameters were assessed to investigate a potential contribution of higher cytokine secretion in female patient hDF on FD sensory-like neuron electrical activity (Fig 4A). Recordings were performed after incubation of iPSC-derived sensory-like neurons with IL-8 or hDF supernatant. Analysis of AP parameters revealed no differences (Fig 4B), however, hyperpolarization amplitudes were higher in Ctrl neurons incubated with Ctrl ($p < 0.001$), male FD hDF supernatant ($p < 0.01$) or IL-8 ($p < 0.001$) compared to naïve Ctrl neurons. The same was true

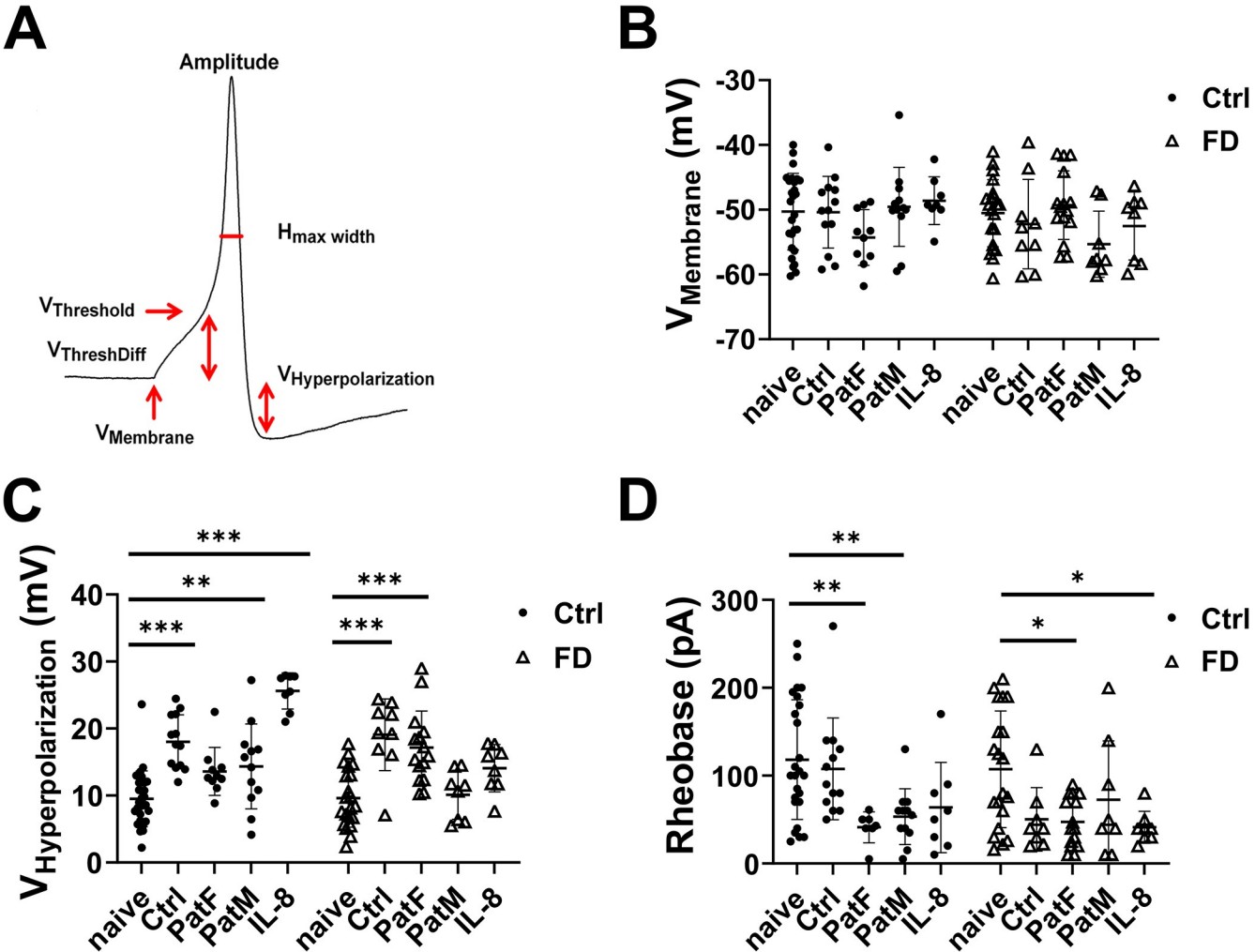

**Fig 4. Electrophysiological analysis of iPSC-derived neurons upon cytokine exposure.** AP parameters (A) were investigated under naïve conditions and after incubation of iPSC-derived sensory-like neurons with IL-8 or hDF supernatant. No difference was found in RMP (B) or any other AP parameter, except for hyperpolarization amplitudes. Hyperpolarization (C) was higher in Ctrl neurons incubated with Ctrl, PatM hDF supernatant, and IL-8 compared to naïve Ctrl neurons ($p < 0.001$; $p < 0.01$; $p < 0.001$). FD neurons revealed increased hyperpolarization amplitudes incubated with Ctrl or PatF supernatant compared to naïve Ctrl neurons ($p < 0.001$; $p < 0.001$). Rheobase currents (D) to elicit first AP were lower in Ctrl neurons after incubation with PatF and PatM supernatant compared to naïve neurons ($p < 0.01$; $p < 0.01$). In FD neurons, incubation with PatF hDF supernatant or IL-8 lead to reduced rheobase currents compared to naïve neurons ($p < 0.05$; $p < 0.05$). Abbreviations: AP = action potential; Ctrl = control; FD = Fabry disease; hDF = human dermal fibroblasts; IL-8 = interleukin 8; iPSC = induced pluripotent stem cell; PatF = female Fabry disease patient; PatM = male Fabry disease patient; RMP = resting membrane potential. *$p < 0.05$, **$p < 0.01$, ***$p < 0.001$.

for FD neurons incubated with Ctrl (p<0.001) or female FD hDF supernatant (p<0.001) compared to naïve Ctrl (Fig 4C).

Next, we investigated rheobase currents to elicit first AP as a potential marker for pain in female FD patients. Current threshold for first AP induction was reduced in Ctrl neurons after incubation with male and female FD hDF supernatant compared to naïve neurons (p<0.01 each). In FD neurons, incubation with female hDF supernatant and IL-8 led to reduced rheobase currents compared to naïve neurons (p<0.05 each; Fig 4D).

## Discussion

We have investigated Gb3-dependent cutaneous mechanisms potentially underlying sex-associated diversity in FD pain. We found higher gene expression of KCNN4 and IL-8 in dermal fibroblasts of women with FD compared to those of men and Ctrl. Additionally, we showed reduction of rheobase currents needed to elicit AP following incubation of iPSC-derived sensory-like neurons with hDF medium of female FD patients or IL-8. These results indicate a potential link between Gb3-dependent ion channel alterations and cutaneous cytokine expression influencing DRG neurons, which may contribute to pain in female FD patients.

FD-associated pain and its underlying mechanisms are incompletely understood. Although Gb3 accumulations in DRG neurons of FD patients and FD animal models [5, 6] point towards a direct link of neuronal sphingolipid deposits and FD-associated pain, other mechanisms cannot be ruled out. This is particularly true for potential Gb3-independent mechanisms which were discovered recently in a zebrafish model of FD [28]. We investigated skin cells, since they are known to be involved in the processing of chemical and thermal stimuli and cutaneous nociception [10, 11, 13]. Surprisingly, we found no Gb3 deposits in the cytoplasm of FD keratinocytes, but in the cell membrane. This result indicates low production of Gb3 which, however, might eventually accumulate in the membrane due to higher turnover time. In support of this notion, a recent publication showed that intracellular deposits can be cleared by enzyme replacement while membrane-bound Gb3 remained unaffected [29]. Further, although Shiga toxin labeling represents a reliable tool to visualize intracellular Gb3 accumulation, accessibility of membranous glycosphingolipids to toxin binding was shown to depend on cholesterol levels [30]. The mere presence of Shiga toxin signal in the membrane of FD keratinocytes does hence not necessarily implicate missing Gb3 in the healthy controls but could arise from an overall FD-related lipid imbalance. Since comparison between fibroblasts and keratinocytes was not possible in our study due to lack of quantification and diversity in cellular labeling, we focused on the investigation of FD fibroblasts compared to control fibroblasts.

Neuronal ion channels, such as Nav 1.7 or TRPV1 are known to play a key role in small fiber-associated pain syndromes [31, 32] and tactile allodynia [33]. We therefore performed gene expression analysis of pain associated ion channels and found a downregulation of TRPV1 and SCN9A expression, while KCNN4 was upregulated in both FD patient groups. Our study design does not allow a functional interpretation of our data. However, in analogy to our previous findings [9], we speculate that alterations in ion channel expression in fibroblasts may be associated with dysregulation of pro- and anti-inflammatory cytokine release, which then may act on skin nociceptors. KCNN4 encoding the SK/IK channel KCa3.1 is associated with modulation of noxious chemical stimuli and tactile allodynia. This calcium-activated potassium channel is also linked with increased cytokine expression in synovial fibroblasts of rheumatoid arthritis patients [34]. KCa3.1 channel inhibition led to reduced secretion of IL-6 and IL-8 [34]. Cytokines are well known for contributing to neuronal hyperexcitability [35] and decreased AP thresholds [36], which may contribute to neuropathic pain.

Further, we recently provided evidence for pro-inflammatory cytokine release from skin fibroblasts of patients with small fiber neuropathy [37]. Specifically, we showed that keratinocytes and fibroblasts contributed differentially to nociceptor degeneration and sensitization in idiopathic small fiber neuropathy and that it was the fibroblasts that may contribute to the latter by increase in IL-6 and IL-8 expression. Given that pain in FD is based on hereditary small fiber neuropathy, our current data add to the evidence supporting the active role of skin cells during the generation and maintenance of pain. Here, only IL-8 gene expression of female FD fibroblasts was upregulated compared to male FD and control fibroblasts. Therefore, we incubated iPSC-derived sensory-like neurons of a healthy control and a male FD patient with IL-8 or supernatant of Ctrl female FD or male FD fibroblasts. Incubation experiments revealed reduced rheobase in control neurons for both, supernatant from male and female fibroblasts compared to naïve neurons. However, in sensory-like neurons generated from FD patients, rheobase was decreased after incubation with supernatant only from female FD fibroblasts and with IL-8.

Reduction of AP rheobase is linked with increased neuronal excitability, potentially contributing to neuropathic pain [38, 39]. Male FD fibroblast supernatant failed to reduce rheobase currents in FD iPSC-derived sensory-like neurons, indicating a sex-associated difference in cytokine composition released by dermal fibroblasts. This result may provide evidence for different underlying pain mechanisms and pain phenotypes of FD patient subpopulations. Based on these findings, we propose a potential mechanism underlying sex-associated pain phenotype as illustrated in Fig 5: mutations in the *GLA* gene and the consecutive enzyme

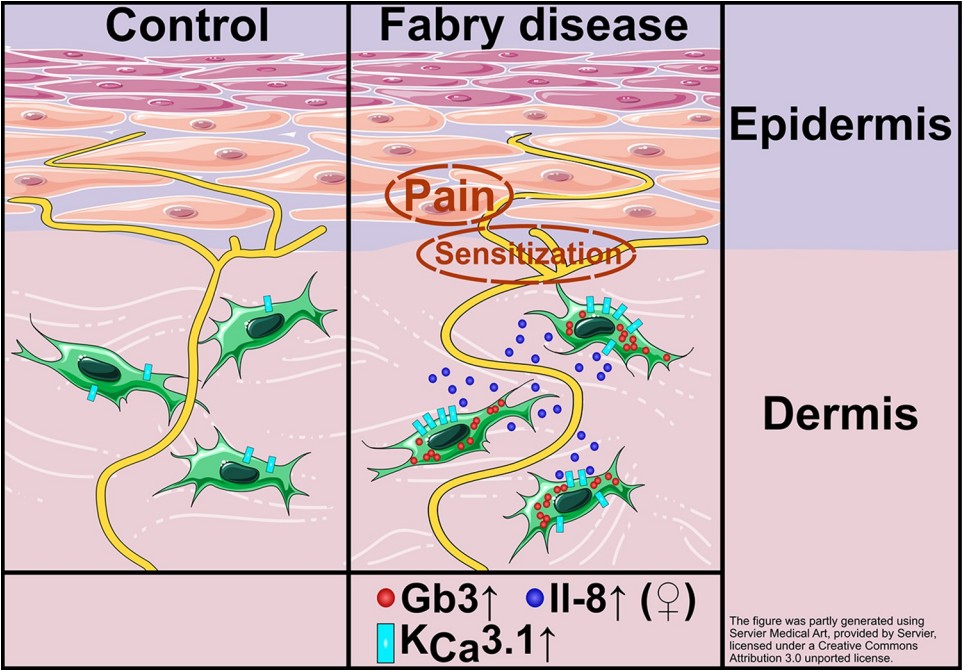

**Fig 5. Potential mechanism underlying sex-associated pain phenotype.** Mutation of the *GLA* gene leads to a non- or dys-functional enzyme, which results in Gb3 accumulation in FD fibroblasts. These deposits may be involved in increased gene expression of the cytokine-linked ion channel KCa3.1. Upregulated ion channel gene expression then may lead to an elevated expression of IL-8 only in female FD patients with a pain phenotype. Potentially increased IL-8 secretion close to nociceptor endings in the skin may then lead to sensitization via reduction of AP induction thresholds contributing to FD-associated pain. Abbreviations: GLA = α-galactosidase A; AP = action potential; FD = Fabry disease; Gb3 = globotriaosylceramide; IL-8 = interleukin 8; KCa3.1 = potassium intermediate/small conductance calcium-activated channel, subfamily N, member 4.

malfunction may lead to Gb3 accumulation in FD fibroblasts. Gb3 depositions may induce increased gene expression of the specific, cytokine-linked ion channel KCa3.1 leading to higher expression of IL-8 only in female FD patients with a pain phenotype. Elevated IL-8 secretion in the vicinity of nociceptor free nerve endings of DRG neurons may then lead to sensitization via reduction of AP induction thresholds contributing to FD-associated pain in women. In men with FD and increased *KCNN4* expression, other cytokines may be at play which needs further exploration.

As for the differences in expression of KCNN4 and IL8 in women and men with FD, we can only speculate that e.g. the individual genetic variant and / or genetic mosaicism in women with FD may have contributed. These factors may also be involved in the diversity of data obtained in the female cohort itself. Further, hormonal influences on skin cells and the secretion of hormones by skin cells may be diverse and together with the genetic background contribute to diversity in ion channel expression and / or cytokine profiles.

Our study has some limitations. Due to the X-linked inheritance, we could not investigate female iPSC-derived sensory-like neurons since iPSC may undergo *in vitro* restitution by skewed X-chromosomal inactivation [25, 40], but had to restrict our experiments to sensory-like neurons obtained from male patients and controls. Further, we limited our experiments to gene expression and cannot rule out that other cytokines or proteins released by skin cells play a role in FD pain. Another limitation of our study is the small sample size, which is due to the non-high-throughput nature of our key experimental methods, as well as the heterogeneity in FD patients. Further the recently discovered fact that Gb3-independent processes may be well at play in the pathophysiology of FD [28] needs to be considered when interpreting our data.

Despite these limitations, our findings provide first insights into potential differences underlying sex-dependent diversity in FD pain phenotypes. A fully human *in vitro* model system was used to investigate the interaction of skin cells and sensory-like neurons in FD-associated pain. Extensive molecular and electrophysiological investigation provided evidence for a potential role of FD-induced skin cell ion channel expression alterations, leading to sensory-like neuron sensitization by cytokine secretion in female FD patients.

## Supporting information

**S1 Table. Patient population.**
(DOCX)

**S1 Data.**
(XLSX)

**S2 Data.**
(XLSX)

## Acknowledgments

Excellent technical help by Daniela Urlaub and Danilo Prtvar is gratefully acknowledged; we also thank Prof. Claudia Sommer for support during patient recruitment (Department of Neurology, University Hospital of Würzburg). The authors further thank the undergraduate students Bettina Vignolo, MSc and Frederik Bär, MSc for their technical contributions to cell culture work (Department of Neurology, University of Würzburg).

## Author Contributions

**Conceptualization:** Lukas Hofmann, Erhard Wischmeyer, Nurcan Üçeyler.

**Data curation:** Julia Grüner, Thomas Klein, Erhard Wischmeyer, Nurcan Üçeyler.

**Formal analysis:** Lukas Hofmann, Julia Grüner, Katharina Klug, Maximilian Breyer, Vanessa Hochheimer, Laura Wagenhäuser, Erhard Wischmeyer, Nurcan Üçeyler.

**Funding acquisition:** Nurcan Üçeyler.

**Investigation:** Lukas Hofmann, Katharina Klug, Maximilian Breyer, Vanessa Hochheimer, Laura Wagenhäuser, Nurcan Üçeyler.

**Methodology:** Lukas Hofmann, Katharina Klug, Maximilian Breyer, Thomas Klein, Vanessa Hochheimer, Laura Wagenhäuser, Erhard Wischmeyer, Nurcan Üçeyler.

**Project administration:** Nurcan Üçeyler.

**Resources:** Erhard Wischmeyer, Nurcan Üçeyler.

**Software:** Nurcan Üçeyler.

**Supervision:** Nurcan Üçeyler.

**Validation:** Nurcan Üçeyler.

**Writing – original draft:** Lukas Hofmann, Nurcan Üçeyler.

**Writing – review & editing:** Lukas Hofmann, Julia Grüner, Katharina Klug, Maximilian Breyer, Thomas Klein, Vanessa Hochheimer, Laura Wagenhäuser, Erhard Wischmeyer, Nurcan Üçeyler.

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
