## [Decision Letter · Decision Letter 0]

20 Nov 2023

PONE-D-23-25958Elevated interleukin-8 expression by skin fibroblasts as a potential contributor to pain in severely affected women with Fabry diseasePLOS ONE

Dear Dr. Ũçeyler,

Thank you for submitting your manuscript to PLOS ONE. After careful consideration, we feel that it has merit but does not fully meet PLOS ONE’s publication criteria as it currently stands. Therefore, we invite you to submit a revised version of the manuscript that addresses the points raised during the review process. The two reviewers have a favorable view of your manuscript. However, they both have extensive and constructive criticisms that you are requested to address as fully as possible. In particular, they feel that data that are referred to are not always fully presented. 

We look forward to receiving your revised manuscript.

Kind regards,

Israel Silman

Academic Editor

PLOS ONE

Journal Requirements:

3. Please amend the manuscript submission data (via Edit Submission) to include author Thomas Klein.

Reviewers' comments:

Reviewer's Responses to Questions

**Comments to the Author**

1. Is the manuscript technically sound, and do the data support the conclusions?

Reviewer #1: Partly

Reviewer #2: Partly

2. Has the statistical analysis been performed appropriately and rigorously? 

Reviewer #1: Yes

Reviewer #2: No

3. Have the authors made all data underlying the findings in their manuscript fully available?

Reviewer #1: Yes

Reviewer #2: Yes

4. Is the manuscript presented in an intelligible fashion and written in standard English?

Reviewer #1: Yes

Reviewer #2: Yes

5. Review Comments to the Author

Reviewer #1: This study presents the results of original research. The study aims at elucidating the link between skin cells and nociceptor sensitization contribution to Fabry Disease (FD) pain in a sex-specific manner.

For doing this, the authors have used cultured keratinocytes and fibroblasts of adult FD patients and healthy controls.

Overall, the manuscript presents comprehensive investigation into the role of IL8 in pain in FD patients.

However, there are some important issues:

Clinical characteristics of the patients and controls are incomplete. I miss more information (e.g. a table with age, sex, age of disease diagnosis, treatment, mutation and alpha-gal activity) and if the controls and FD patients were matched in any way.

The authors stated that GB3 deposits were detected in control hDF. Do you have any hypothesis of why? Was this deposition present in all the control samples?

In figure 1, only one sample of each condition is represented. Could you include the quantification of all the samples to give a better overview of the differences?

In figures 2 and 3, I guess each point represents a patient/control. Did you perform technical replicates? Are you using the mean for each technical replicate? Can you include the mean±SD for each sample?

Also, in figures 2 and 3: are the figure outliers included in the statistical test? Do you have any hypothesis/data about the patient that can explain the huge difference in, for example, between one of the samples in fig. 2C (outlier) and the rest of the group (FD female)?

Do you have any hypothesis of why the differences between in KCNN4 and IL8 in women and men? (Discussion, 284-286)

Could you discuss why GB3 deposits are found in the cell membrane but not in the cytoplasm? (Discussion, line 296)

In relation to the clinical data, have you check differences in IL8, KCNN4, etc depending on age at which pain appeared?

In limitations, I would add human heterogeneity and sample size.

In general, the experiments are not described in sufficient detail and discussion must be improved. Also, the conclusion must be improved, as includes methods.

Reviewer #2: Please see attached file.

6. PLOS authors have the option to publish the peer review history of their article (what does this mean?). If published, this will include your full peer review and any attached files.

Reviewer #1: No

Reviewer #2: No

---

## [Author Response · Author response to Decision Letter 0]

26 Jan 2024

Von: em.pone.0.878373.9afdce88@editorialmanager.com <em.pone.0.878373.9afdce88@editorialmanager.com> Im Auftrag von PLOS ONE

Gesendet: Dienstag, 21. November 2023 00:21

An: Üceyler, Nurcan <Ueceyler_N@ukw.de>

Betreff: [EXT] PLOS ONE Decision: Revision required [PONE-D-23-25958] - [EMID:9d6f74c98acca272]

PONE-D-23-25958

Elevated interleukin-8 expression by skin fibroblasts as a potential contributor to pain in severely affected women with Fabry disease

PLOS ONE

Dear Dr. Ũçeyler,

Thank you for submitting your manuscript to PLOS ONE. After careful consideration, we feel that it has merit but does not fully meet PLOS ONE’s publication criteria as it currently stands. Therefore, we invite you to submit a revised version of the manuscript that addresses the points raised during the review process.

The two reviewers have a favorable view of your manuscript. However, they both have extensive and constructive criticisms that you are requested to address as fully as possible. In particular, they feel that data that are referred to are not always fully presented. 

We look forward to receiving your revised manuscript.

Kind regards,

Israel Silman

Academic Editor

PLOS ONE

Journal Requirements:

Done. 

Thank you for the information. 

3. Please amend the manuscript submission data (via Edit Submission) to include author Thomas Klein.

Done. 

We show all data obtained and have eliminated “data not shown”.  

Reviewers' comments:

Reviewer's Responses to Questions

Comments to the Author

1. Is the manuscript technically sound, and do the data support the conclusions?

Reviewer #1: Partly

Reviewer #2: Partly

2. Has the statistical analysis been performed appropriately and rigorously? 

Reviewer #1: Yes

Reviewer #2: No

3. Have the authors made all data underlying the findings in their manuscript fully available?

Reviewer #1: Yes

Reviewer #2: Yes

4. Is the manuscript presented in an intelligible fashion and written in standard English?

Reviewer #1: Yes

Reviewer #2: Yes

5. Review Comments to the Author

Reviewer #1: 

This study presents the results of original research. The study aims at elucidating the link between skin cells and nociceptor sensitization contribution to Fabry Disease (FD) pain in a sex-specific manner. For doing this, the authors have used cultured keratinocytes and fibroblasts of adult FD patients and healthy controls. Overall, the manuscript presents comprehensive investigation into the role of IL8 in pain in FD patients. However, there are some important issues:

Clinical characteristics of the patients and controls are incomplete. I miss more information (e.g. a table with age, sex, age of disease diagnosis, treatment, mutation and alpha-gal activity) and if the controls and FD patients were matched in any way.

We have included supplementary Table 1 with the requested data in our revised manuscript. 

The authors stated that GB3 deposits were detected in control hDF. Do you have any hypothesis of why? Was this deposition present in all the control samples?

Gb3 is a physiologically produced sphingolipid in human cells, hence, it is not surprising but indeed expected to observe low amounts of Gb3 in control samples. We have added this information in the Results section of our revised manuscript (please see page 12): 

“As expected, only few Gb3 deposits were detected in control hDF reflecting its physiological generation (Fig. 1C`).”

In figure 1, only one sample of each condition is represented. Could you include the quantification of all the samples to give a better overview of the differences?

We have not performed a quantification on this occasion since we merely aimed at qualitatively demonstrating the obvious difference between Gb3-positive and -negative cells. We have included this aspect in our revised manuscript (please see page 12): 

“We first investigated FD and control cells for qualitative comparison.” 

In figures 2 and 3, I guess each point represents a patient/control. Did you perform technical replicates? Are you using the mean for each technical replicate? Can you include the mean±SD for each sample? 

We performed n=3 technical replicates on the same 96-well plate and we calculated the mean of the Ct values. For comparative data analysis, we applied the delta-delta-Ct method as described in our original manuscript. Hence, we cannot include the mean±SD which would also make Fig. 2 and 3 unreadable. However, we provide the raw data of our qPCR runs as supplementary file referred to in the revised Legends of Fig. 2 and 3. 

Also, in figures 2 and 3: are the figure outliers included in the statistical test? Do you have any hypothesis/data about the patient that can explain the huge difference in, for example, between one of the samples in fig. 2C (outlier) and the rest of the group (FD female)?

We refrained from eliminating “outliers” from the statistical tests, but used all data obtained. We have carefully checked and controlled quality standards of the biomaterial used and also patients` clinical data, however, we could not find a potential reason for the variability. At this stage, we would refrain from mere speculative hypotheses on why this might be, but would rather like to show the data as is. 

Do you have any hypothesis of why the differences between in KCNN4 and IL8 in women and men? (Discussion, 284-286). 

Our study design does not allow functional conclusions, hence, we can only speculate that e.g. genetic mosaicism in women with FD may contribute to the differences found between sexes. This may also be involved in the diversity of data obtained in the female cohort itself. Of course, hormonal influences on skin cells and the secretion of hormones by skin cells may also be diverse and together with the genetic background contribute to variability in ion channel expression and / or cytokine profiles. We have included a respective passage in the revised Discussion of our manuscript (please see page 17): 

“As for the differences in expression of KCNN4 and IL8 in women and men with FD, we can only speculate that e.g. the individual genetic variant and / or genetic mosaicism in women with FD may have contributed. These factors may also be involved in the diversity of data obtained in the female cohort itself. Further, hormonal influences on skin cells and the secretion of hormones by skin cells may be diverse and together with the genetic background contribute to diversity in ion channel expression and / or cytokine profiles.”

Could you discuss why GB3 deposits are found in the cell membrane but not in the cytoplasm? (Discussion, line 296).

Thank you for this comment. We have included the following passage in the Discussion section of our revised manuscript (please see page 15): 

“Surprisingly, we found no Gb3 deposits in the cytoplasm of FD keratinocytes, but in the cell membrane. This result indicates low production of Gb3 which, however, might eventually accumulate in the membrane due to higher turnover time. In support of this notion, a recent publication showed that intracellular deposits can be cleared by enzyme replacement while membrane-bound Gb3 remained unaffected [34]. Further, although Shiga toxin labeling represents a reliable tool to visualize intracellular Gb3 accumulation, accessibility of membranous glycosphingolipids to toxin binding was shown to depend on cholesterol levels [35]. The mere presence of Shiga toxin signal in the membrane of FD keratinocytes does hence not necessarily implicate missing Gb3 in the healthy controls but could arise from an overall FD-related lipid imbalance."

In relation to the clinical data, have you check differences in IL8, KCNN4, etc depending on age at which pain appeared?

Unfortunately, we cannot perform this correlation analysis, since our patients could only provide rough estimations on when pain first appeared, please see data in the new supplementary Table 1. 

In limitations, I would add human heterogeneity and sample size.

We have added these aspects to the revised Limitation section of our manuscript (please see page 18). 

“Another limitation of our study is the small sample size, which is due to the non-high-throughput nature of our key experimental methods, as well as the heterogeneity in FD patients.“

In general, the experiments are not described in sufficient detail and discussion must be improved. Also, the conclusion must be improved, as includes methods.

Please also see our replies to respective questions of Reviewer 2 below. We have revised our manuscript accordingly. Additionally, we have included further details within the patch-clamp section covering recordings of action potentials, about the supernatant medium, and patients` pain phenotype in the revised Methods section of our manuscript. We have also improved our revised Discussion (please see respective passages in tracked mode in the text).

Reviewer #2: Please see attached file.

Hofmann, et. al. present this manuscript aimed at defining molecular mechanisms in dermal fibroblasts that contribute to pain in Fabry disease. Fabry disease is a lysosomal storage disease arising from X-linked loss-of-function mutations in α-galactosidase A (GLA) that frequently results in a small fiber neuropathy and chronic pain the distal extremities. Substantial evidence in the literature demonstrates accumulation of globotriaosylceramide (Gb3), the subtracted for α-galactosidase A, in the dorsal root ganglia contributes to nociception and pain in Fabry disease; however, recent evidence highlights the role for non-neuronal cell types in pain generally (e.g., keratinocytes, fibroblasts, leukocytes, Schwann cells) and in Fabry disease specifically (e.g., Schwann cells, possibly fibroblasts). To determine which cutaneous cell types may contribute to pain in Fabry disease, the authors recruited a cohort male and female patients with Fabry disease and a cohort of controls from their unaffected family members and friends, cultured and characterized keratinocytes and dermal fibroblasts from these cohorts, and assessed the effects of conditioned media from dermal fibroblasts or interleukin-8 (IL-8) on intrinsic excitability of induced pluripotent stem cells (iPSCs) differentiated into sensory neuron-like cells. The authors observe increased Gb3 accumulation in fibroblasts, but not keratinocytes, from Fabry disease patients, and, thus, focus the reset of their analyses on dermal fibroblasts. The authors also find decreased expression of TRPV1 and NaV1.7 and increased expression of KCNN4 and Il-8 in dermal fibroblasts derived from Fabry disease. The authors also report reduced rheobase (current threshold to evoke a single action potential) in iPSC-derived sensory-like neurons derived from control and a male patient with Fabry disease exposed to conditioned media from dermal fibroblasts derived from a female Fabry disease patient or IL8. From this, the authors conclude that Gb3 accumulation in dermal fibroblasts alters ion channel and cytokine expression, thereby sensitizing nociceptors and contributing to pain in Fabry disease in a sex-specific manner. This study is quite important and valuable because it is extremely difficult to get tissue samples from Fabry patients and here the authors have conducted a body of work on these patient tissues. Very few, if any other labs, could obtain these tissues and perform these studies; these studies are critical for translating findings in rodent models of Fabry disease. The idea that dermal fibroblasts could contribute to nociception and pain in Fabry disease is meritorious and warrants further study. Overall, I like this study a lot and think this is an appropriate venue for this study. That said, there are several issues with the data and interpretation that I think should be addressed before publication to make a very robust, impactful publication. 

Major revisions:

1) Not all data reported are presented. Specifically:

a) Patient demographics are not presented in the manuscript.

We have added patient demographics in the new supplementary Table 1 of our revised manuscript. 

b) IENF density measurements are referred to in line 233-234, yet not presented anywhere in the manuscript. Given how these data are described in this section, there may be issues with IENFD analysis as well (see below).

We provide distal and proximal IENFD in the new supplementary Table 1 of our revised manuscript. 

c) In line 249, the authors write that “NaV1.8 and TRPM8 were not expressed (data not shown).” The authors should present this in their figures.

Thank you for the opportunity to clarify this point. The Ct values found for Nav1.8 and TRPM8 were >33, indicating that both genes are hardly expressed. This can, unfortunately, not be visualized in the Figures in a meaningful manner. However, we have followed the Reviewer`s suggestion and have included the Ct values for clarification in the Results section of our revised manuscript (please see page 13): 

“SCN10A and TRPM8 were not expressed (Ct values >33 each).”

d) Figure 4 presents electrophysiological analysis of iPSC-derived sensory-like neurons dosed with either IL-8 or dermal fibroblast conditioned media from controls, male Fabry disease patients, or female Fabry disease patients. However, the current figure only presents panels for action potential amplitude or rheobase from iPSC-derived sensory-like neurons incubated with conditioned media from control or female patient-derived dermal fibroblasts. Moreover, the authors refer to Figure 4C in line 280, yet there is no C panel presented in Figure 4.

We apologize for this oversight and have re-uploaded Fig. 4 including 4C. 

2) The results as presented lack appropriate controls for sex as a biological variable. The authors are right in looking at sex as a potential variable in patients with Fabry disease, but there is no account for potential effects of sex in control fibroblasts. There is substantial rationale that sex in unaffected, healthy patients affects gene expression and release of factors into media. Because sex is not treated as an independent variable, the authors should not claim that the results here are an interaction of sex and disease state (e.g., a sexually dimorphic factor of Fabry disease) or simply additive main effects of sex and disease.

We apologize for the confusion which may have appeared due to the erroneously uploaded old version of Fig. 4. We have now re-uploaded the correct version of Figure 4. Please note that we have indeed used control media of men and women. 

a) The point above also applies to the sex of iPSCs used in these studies. The authors provide strong rationale for only using iPSC-derived sensory-like neurons from a male Fabry disease patient due to availability, yet in the same preprint where they first describe the lines used herein, they developed a female Fabry patient iPSC-derived sensory neuron-like line (FD-3). The use of only male Fabry disease iPSC lines requires stronger justification. Further, the sex of the control iPSC line is not clearly indicated. 

As we have shown in our preprint (Klein et al., 2023; meanwhile manuscript provisionally accepted by Brain Communications), iPSC obtained from men with FD develop persisting Gb3 accumulations, while iPSC derived from our female patient underwent in vitro restitution by skewed X-chromosomal inactivation. Hence, these iPSC do not show any cellular hints for FD, particularly no Gb3 depositions. Please note that we therefore did not further differentiated iPSC of the female FD patient to sensory-like neurons as described in the respective preprint (Klein et al., 2023; meanwhile manuscript provisionally accepted by Brain Communications). We have included this information in the revised Discussion of our manuscript (please see page 18): 

“Due to the X-linked inheritance, we could not investigate female iPSC-derived sensory-like neurons since iPSC may undergo in vitro restitution by skewed X-chromosomal inactivation [26,41],…”

Sex of our healthy control line was male, which we have added in our revised manuscript (please see page 10). 

3) There is insufficient detail in the Experimental Methods section:

a) In describing quantification of IENF density, the authors are right in citing Lauria, et. al. (2010); however, they should briefly describe how IENFs were quantified herein instead of writing “Intraepidermal nerve fibers were quantified according to published rules,” as various methods of IENF quantification have been published, and these variations in quantification can lead to different results.

We have added further details as for the IENF counting in the revised Methods of our manuscript (please see page 5):

“Intraepidermal nerve fibers were quantified according to published rules [18] and the mean intraepidermal nerve fiber density (i.e. the number of fibers/mm) was determined by counting intraepidermal nerve fibers, if they crossed or originated at the dermal-epidermal border. The number of fibers was divided by the length of the respective epidermis. Secondary branches and fiber fragments were spared.“

i) The authors must clarify how these data were analyzed. In their results (lines 223-234), the authors write that only a proportion of patients demonstrated reduced IENFD. If this was determined relative to the control cohort presented in this paper (this appears likely given the methods section detailing use of skin punch biopsies) this analysis is not standard and may be inappropriate. Instead, the authors should compare IENFD between patient and control cohorts by the appropriate parametric or nonparametric test (include representative images and graphical presentation). If the authors instead compared IENFD of patients to age- and sex-adjusted normative values, the authors should explicitly state and cite the source of those values and include a description of how experimental values were determined to be reduced relative to those normative values.

We have compared our patient IENFD to our laboratory control data, which is also used for our clinical routine diagnostics. This normative data base grounds on IENFD data of 56 healthy men and 124 healthy women. We have included respective details in the revised Methods of our revised manuscript (please see page 5): 

“IENFD data were compared with our laboratory`s normative data base grounding on 180 healthy controls (124 women; median age: 50 years, 20–84; 56 men; median age 53 years, 22–76) in skin biopsies from the lower leg (women: n = 109; men: n = 46) and the upper thigh (women: n = 102; men: n = 31). All skin biopsies were processed and assessed in our laboratory.” 

b) It is unclear how many passages of human-derived keratinocytes and dermal fibroblasts were used in these studies. This should be included in the cell culture section of the methods.

We used maximum two passages after thawing. We have included respective information in our revised manuscript (please see page 6): 

“We used maximum two passages after thawing.”

c) The gene expression analysis section of the methods requires the additional detail. As presented, replication of these methods would not be possible.

i) It is unclear from the methods whether extracted RNA was quantified or assessed for purity and integrity by UV spectrophotometry (e.g., by a NanoDrop, which is standard in the field).

We have used a NanoDrop and have included respective information in our revised manuscript (please see page 8): 

“We used the NanoDrop™ One (Thermo Fisher Scientific, Waltham, MA, USA) for assessment of RNA quality and quantity.” 

ii) The authors do not indicate that they treated their RNA samples with DNase, which is essential to avoid amplifying genomic DNA in their qPCR.

DNase treatment is a standard step within the used RNAesy kit which we have used. We have highlighted respective information in our revised Methods (please see page 8):

“Applying miRNeasy Mini Kit (Qiagen, Hilden, Germany) and a Centrifuge 5417R (Eppendorf, Hamburg, Germany), we extracted mRNA from lysed cells including DNAse treatment.”

iii) It is not clear how much mRNA was used in the reverse transcriptase reaction, nor do the methods state how much cDNA was loaded for the qPCR reaction.

We used 250 ng of mRNA and 3.5 µl of cDNA which we have included in our revised Methods (please see page 8): 

“Applying TaqMan Reverse Transcription reagents, 250 ng of mRNA was transcribed to cDNA. Mastermix contained 5 μl random hexamer, 10 μl 10x buffer, 22 μl MgCl2, 20 μl dNTP, 2 μl RNAse inhibitor, 6.2 μl multiscribe RT, and 2 μl Oligo-D and 3.5 µl of cDNA was individually added.”

iv) Primers or probe sequences available in a supplementary table.

We have used TaqMan probes and Thermo Fisher Scientific does not provide sequence information, unfortunately. The respective AssayIDs were provided in the original manuscript. 

d) The authors description of the generation and characterization of iPSC-derived sensory-like neurons in this manuscript is insufficient and the following should be addressed:

i) The description of the generating these cell lines is cited, but the specific citation for generating control, FD-1 and FD-2 cell lines is a preprint that has yet to undergo peer review. As such, the use of these cells here warrants a more in-depth description of the generation and characterization of these cells. Additionally, FD-2 (a Fabry disease sensory neuron-like line derived from a patient without pain) is not used in this manuscript. In the initial preprint cited here, the FD-2 and control lines were generated with the reprogramming cocktail over the same time course, while FD-1 used a shorter timecourse.

Klein et al., 2023 is published as a preprint, meanwhile provisionally accepted for publication in Brain Communications, and all information on the generation of our sensory-like neurons are described therein in extensive detail. The Reviewer will understand that the extent of cellular characterization would go far beyond the scope of our current manuscript and we would also not like to generate a “double publication” with all the information already provided in our preprint Klein et al., 2023 and now also provisionally accepted manuscript. Our sensory-like neurons were generated following the protocol by Eberhardt et al., 2015 (Eberhardt E, Havlicek S, Schmidt D, et al. Pattern of Functional TTX-Resistant Sodium Channels Reveals a Developmental Stage of Human iPSC- and ESC-Derived Nociceptors. Stem Cell Reports. Sep 8 2015;5(3):305-13. doi:10.1016/j.stemcr.2015.07.010) which is based on the original publication by Chambers et al., 2009 (Chambers SM, Fasano CA, Papapetrou EP, Tomishima M, Sadelain M, Studer L. Highly efficient neural conversion of human ES and iPS cells by dual inhibition of SMAD signaling. Nat Biotechnol. Mar 2009;27(3):275-80. doi:10.1038/nbt.1529). We have included the latter citation in our revised Methods. 

Meanwhile, we have published another peer-reviewed manuscript also giving further details on the generation of our sensory-like neurons (Breyer et al., Mol Genet Metab Rep, 2023). NB: as for our preprint Klein et al.: in case of acceptance of our current manuscript, we will be happy to update the citation for Klein et al. in the proofs, respectively since we expect to receive the final acceptance in the meantime. 

As for the question on “a shorter time course”: All lines used in this manuscript (FD-1, FD-2, Ctrl) were reprogrammed using the StemRNA 3rd Gen reprogramming kit (Reprocell) with the same time frame for the reprogramming. The only line (not of relevance for this study) that was reprogrammed with a different time frame was “FD-3”. This line was reprogrammed using the “StemMACS mRNA reprogramming (Miltenyi Biotec). Despite the name, this was the first line reprogrammed in our lab, but the kit was discontinued, so we had to switch suppliers. The general mechanism of reprogramming stays the same and more importantly, all cells passed the necessary pluripotency tests. The needed time frame can differ from kit to kit, due to varying concentration, mRNA half-life etc. We followed the manufacturer’s instructions for both kits.

ii) The authors initially refer to these cell lines as “nociceptive neurons” (line 182-184). It isn’t clear that these are nociceptors. Through the rest of the manuscript, the authors refer to these lines as “sensory neurons”, which is more accurate.

Thank you, we have changed the wording to “sensory-like neurons” throughout the text. 

e) In the patch-clamp analysis section of their methods, the authors do not indicate how many cells were recorded for each media condition.

We have recorded ≥6 cells per each media condition and have included this information in our revised Methods (please see page 11):

“For all patch-clamp experiments and all conditions investigated, ≥6 cells were recorded.”

4) The authors write in the “Statistical analysis” section of the Methods that the Kruskal-Wallis test was used for non-parametric data (followed by Dunn’s multiple comparisons as a post hoc, both gene expression and electrophysiological analyses). As no statistical tests are presented for parametric data in this section, it is assumed that Kruskal-Wallis is the main test used herein. The Kruskal-Wallis test is appropriate for nonparametric analyses with a single independent variable (akin to a one-way ANOVA for parametric data). Notwithstanding including sex as an independent variable throughout the manuscript (see Major Revision 2), Figure 4A-B present electrophysiological data with at least two independent variables (e.g., control vs Fabry disease iPSC-derived sensory-like neurons and media treatments), rendering the Kruskal-Wallis test inappropriate. In all cases where the data are nonparametric and include more than a single independent variable, the authors should use the Friedmann’s test. If the data are parametric, the authors should use a 2- or 3-way ANOVA as appropriate.

We thank the Reviewer for raising this point. We agree that the statistical analysis of our electrophysiological data includes two independent variables (“cell line” and “treatment”). Hence, we have revised the analysis and have applied two-way ANOVAs on the dependent variables “VMembrane”, “VHyperpolarization”, and “Rheobase”. We have included this information in the Methods of our revised manuscript (please see page 11):

“Normally distributed electrophysiological parameters were analyzed using two-way ANOVAs with cell line (Ctrl, FD) and treatment (naïve, Ctrl, PatF, PatM, IL-8) as independent variables. For multiple comparison, Dunnett’s test was applied.”

5) The authors conclude that the mechanisms they uncover are Gb3-dependent, but this is not necessarily true. Recent evidence suggests Gb3-independent roles of Gla-deficiency in a zebrafish model of Fabry disease (Elsaid, et. al., 2023, J Trans Med). The authors must at least discuss the possibility of Gb3-independent mechanisms driving the changes observed herein.

Thank you for pointing this out. We have included Elsaid et al., 2023 and a respective statement in the revised Discussion of our manuscript (please see page 15):

“This is particularly true for potential Gb3-independent mechanisms which were discovered recently in a zebrafish model of FD [29].”

a) Additionally, in this analysis, the authors should consider that keratinocytes, even though they may not accumulate Gb3 in culture, may still play a role in increased pain and nociception in Fabry disease through Gb3-independent consequences of GLA-deficiency.

We have likewise included this aspect in the revised Discussion of our manuscript (please see page 18): 

“Further the recently discovered fact that Gb3-independent processes may be well at play in the pathophysiology of FD [29] needs to be considered when interpreting our data.”

6) The authors could provide a stronger explanation for why changes in ion channel expression—and especially decreased expression of sodium and calcium channels with concomitant increase in potassium channels—in fibroblasts could be a mechanism driving nociceptor sensitization or pain. The specific changes seen herein suggest that fibroblasts might be less excitable. Why would this lead to increased neuronal excitability? As it stands, the expression data for these channels is purely correlative.

Our study design and the low number of samples does not allow a robust functional interpretation of our data, unfortunately. In analogy to our previous findings (Rickert et al, 2020), we speculate that alterations in ion channel expression in fibroblasts, as non-neuronal cells, may be associated e.g. with dysregulation of pro- and anti-inflammatory cytokine release, which then may act on skin nociceptors. We have included a respective passage in our revised Discussion (please see page 16): 

“Our study design does not allow a functional interpretation of our data. However, in analogy to our previous findings [9], we speculate that alterations in ion channel expression in fibroblasts may be associated with dysregulation of pro- and anti-inflammatory cytokine release, which then may act on skin nociceptors.”

7) In the title and throughout the manuscript, the authors describe their proposed mechanism as a driver of pain in severely affected women with Fabry disease. But the authors do not present any data on pain in their cohorts nor criterion for considering a patient’s pain severe. This needs to be remedied—either by including such data and criterion, or by softening the language. 

We have followed the Reviewer`s suggestion and have eliminated the expression of “severely affected” in our revised manuscript and have included data on pain severity in the new supplementary Table 1. 

Minor revisions:

1) For PCR analyses, the authors should use the appropriate gene names (e.g., NaV1.7 is SCN9A, you are not looking at the protein product of that gene, but it’s transcript). As a more central example, the gene abbreviation of α-galactosidase A for humans is not α-GAL, but GLA.

We have edited the respective passages. 

a) Conversely, at some point in the discussion, the authors should mention that KCNN4 encodes the SK/IK channel. Please do not remove all mention of the functional protein product of these genes (NaV1.7); rather, when talking about gene expression, use the gene name. When talking about the channel, refer to the protein name. This will improve readability and accessibility.

We have edited the text respectively. 

2) While the authors never directly compare the accumulation of Gb3 in dermal fibroblasts to keratinocytes, they should highlight that this comparison is not possible from their current data. There are two reasons for this:

a) Accumulation of Gb3 was not quantified between these cell types.

b) The STX used to stain for Gb3 in keratinocytes and fibroblasts cannot be directly compared, as they do not use the same fluorophore.

We have followed the Reviewer`s suggestion and have added a respective statement in the revised Methods (please see page 15): 

“Since comparison between fibroblasts and keratinocytes was not possible in our study due to lack of quantification and diversity in cellular labeling, we focused on the investigation of FD fibroblasts compared to control fibroblasts.”

3) The authors should refine their language in the second paragraph of their patch-clamp analysis section (lines 203-211) to more clearly indicate that all patient conditioned media samples were used as individual replicates.

We have added a respective statement in the revised Methods (please see page 11): 

“Supernatants were not pooled and all patient conditioned media samples were used as individual replicates.”

4) The authors should better cite key contributors to the notion that several types of cutaneous cells are known contributors to nociception and somatosensation. Only one of the cited works on fibroblasts (Kreß, et. al. (2022), PAIN) directly shows any role for fibroblasts in neuronal excitability. The role for fibroblast control of nociception or pain has not been cited.

We have followed the Reviewer`s suggestion and have extended our citations respectively (please see page 3): 

“Further research underscores the role of fibroblast control of nociception in animal and in vitro studies [18-20].”

5) The Summary Figure 5 is very crudely drawn and should be made more professional. Additionally, it is not immediately clear what the figure is trying to convey, and clarity could be improved with arrows indicating the flow of ideas.

We have revised Figure 5 accordingly.

6) The senior author has previously published that fibroblast-derived conditioned media from patients with small fiber neuropathy sensitizes nociceptors. The authors should better place the results of their current manuscript in the context of their previous findings. This will strengthen the paper.

We have included a respective passage in the revised Discussion (please see page 16): 

“Further, we recently provided evidence for pro-inflammatory cytokine release from skin fibroblasts of patients with small fiber neuropathy [38]. Specifically, we showed that keratinocytes and fibroblasts contributed differentially to nociceptor degeneration and sensitization in idiopathic small fiber neuropathy and that it was the fibroblasts that may contribute to the latter by increase in IL-6 and IL-8 expression. Given that pain in FD is based on hereditary small fiber neuropathy, our current data add to the evidence supporting the active role of skin cells during the generation and maintenance of pain.”

7) In Figure 4B, the data indicate that even control-derived dermal fibroblast conditioned media decreases the rheobase of Fabry patient iPSC-derived sensory-like neurons. In fact, this result is not significantly different from the decrease in rheobase evoked from female patient-derived dermal fibroblast conditioned media. The authors should comment on this in their discussion.

Thank you for pointing this out. With the revised statistical analysis as suggested by the Reviewer, the control-derived fibroblast conditioned media does not have any effect on the rheobase of Fabry patient iPSC-derived sensory-like neurons, please see revised Fig. 4. 

8) Was conditioned media collected in serum-containing or serum-free media? This can have substantial effects on the stability of factors in the media, and the results of placing these conditioned media on other cell types.

Conditioned media was collected in serum-free media. We have added this information in the revised Methods of our manuscript (please see page 11): 

“HDF supernatant was collected 24h after replacement of hDF medium with pure (serum-free) DMEM F12 from 80% confluent fibroblasts.”

9) The authors should read through the current manuscript again for grammatical errors, run-on sentences, and clarity.

Thank you, we have carefully revised and edited all parts of our manuscript. 

6. PLOS authors have the option to publish the peer review history of their article (what does this mean?). If published, this will include your full peer review and any attached files.

Do you want your identity to be public for this peer review? For information about this choice, including consent withdrawal, please see our Privacy Policy.

Reviewer #1: No

Reviewer #2: No

---

## [Decision Letter · Decision Letter 1]

4 Mar 2024

Elevated interleukin-8 expression by skin fibroblasts as a potential contributor to pain in women with Fabry disease

PONE-D-23-25958R1

Dear Dr.Üçeyler,

We’re pleased to inform you that your manuscript has been judged scientifically suitable for publication and will be formally accepted for publication once it meets all outstanding technical requirements.

Kind regards,

Israel Silman

Academic Editor

PLOS ONE

Additional Editor Comments (optional):

Reviewers' comments:

Reviewer's Responses to Questions

**Comments to the Author**

1. If the authors have adequately addressed your comments raised in a previous round of review and you feel that this manuscript is now acceptable for publication, you may indicate that here to bypass the “Comments to the Author” section, enter your conflict of interest statement in the “Confidential to Editor” section, and submit your "Accept" recommendation.

Reviewer #1: All comments have been addressed

Reviewer #2: All comments have been addressed

2. Is the manuscript technically sound, and do the data support the conclusions?

Reviewer #1: Yes

Reviewer #2: Yes

3. Has the statistical analysis been performed appropriately and rigorously? 

Reviewer #1: Yes

Reviewer #2: Yes

4. Have the authors made all data underlying the findings in their manuscript fully available?

Reviewer #1: Yes

Reviewer #2: Yes

5. Is the manuscript presented in an intelligible fashion and written in standard English?

Reviewer #1: Yes

Reviewer #2: Yes

6. Review Comments to the Author

Reviewer #1: The main issues with the the article “Elevated interleukin-8 expression by skin fibroblasts as a potential contributor to pain in women with Fabry disease” have been addressed, and modifications made in line with the suggestions. This modifications have significantly improved the quality and clarity of the manuscript. In conclusion, the revision have significantly strengthened the manuscript, addressing concerns regarding clarity, depth of analysis, and methodological rigor.

Reviewer #2: The authors have done a good job of addressing all of the reviewers’ comments and have sufficiently revised the manuscript. I have no further suggestions to make.

7. PLOS authors have the option to publish the peer review history of their article (what does this mean?). If published, this will include your full peer review and any attached files.

Reviewer #1: No

Reviewer #2: No

---

## [Editor Report · Acceptance letter]

30 Mar 2024

PONE-D-23-25958R1 

PLOS ONE

Dear Dr. Üçeyler, 

I'm pleased to inform you that your manuscript has been deemed suitable for publication in PLOS ONE. Congratulations! Your manuscript is now being handed over to our production team.

Kind regards, 

on behalf of

Prof. Israel Silman 

Academic Editor

PLOS ONE